# Comparative Analysis of the Alkaline Proteolytic Enzymes of *Yarrowia* Clade Species and Their Putative Applications

**DOI:** 10.3390/ijms24076514

**Published:** 2023-03-30

**Authors:** Dominika Ciurko, Cécile Neuvéglise, Maciej Szwechłowicz, Zbigniew Lazar, Tomasz Janek

**Affiliations:** 1Department of Biotechnology and Food Microbiology, Wrocław University of Environmental and Life Sciences, 51-630 Wrocław, Poland; 2SPO, Univ Montpellier, INRAE, Institut Agro, 34060 Montpellier, France

**Keywords:** *Yarrowia* clade, yeast, phylogenetics, protease, brewers’ spent grain

## Abstract

Proteolytic enzymes are commercially valuable and have multiple applications in various industrial sectors. The most studied proteolytic enzymes produced by *Yarrowia lipolytica*, extracellular alkaline protease (Aep) and extracellular acid protease (Axp), were shown to be good candidates for different biotechnological applications. In this study, we performed a comprehensive analysis of the alkaline proteolytic enzymes of *Yarrowia* clade species, including phylogenetic studies, synteny analysis, and protease production and application. Using a combination of comparative genomics approaches based on sequence similarity, synteny conservation, and phylogeny, we reconstructed the evolutionary scenario of the *XPR2* gene for species of the *Yarrowia* clade. Furthermore, except for the proteolytic activity of the analyzed *Yarrowia* clade strains, the brewers’ spent grain (BSG) was used as a substrate to obtain protein hydrolysates with antioxidant activity. For each culture, the degree of hydrolysis was calculated. The most efficient protein hydrolysis was observed in the cultures of *Y. lipolytica*, *Y. galli*, and *Y. alimentaria.* In contrast, the best results obtained using the 2,2-azinobis (3-ethyl-benzothiazoline-6-sulfonic acid (ABTS) method were observed for the culture medium after the growth of *Y. divulgata*, *Y. galli*, and *Y. lipolytica* on BSG.

## 1. Introduction

Nearly two decades ago, the genus *Yarrowia* was still limited to *Yarrowia lipolytica* species [1,2]. In 2004, Péter et al. [3] described *Candida galli*, a sister species of *Y. lipolytica*, which was later renamed *Y. galli.* In 2005, Cletus Kurtzman [4] defined the *Yarrowia* clade as a taxonomic group, including *Y. lipolytica* and *C. galli*, as well as *Candida hispaniensis* and *Candida bentonensis.* Since then, numerous species have been assigned to the *Yarrowia* clade primarily based on marker sequencing, which has become a common tool in taxonomic studies. Today, the *Yarrowia* clade comprises *Y. lipolytica*, *Yarrowia alimentaria*, *Yarrowia bubula*, *Yarrowia brassicae*, *Yarrowia deformans*, *Yarrowia divulgata*, *Y. galli*, *Yarrowia hollandica*, *Yarrowia keelungensis*, *Yarrowia osloensis*, *Yarrowia parophonii*, *Yarrowia phangngaensis*, *Yarrowia porcina*, *Yarrowia yakushimensis*, and *C. hispaniensis* [5]. The species listed above share many morphological and physiological characteristics of *Y. lipolytica.* One is the ability to exist as yeast cells, pseudomycelium, or true septate hyphae. However, the best known feature of all the species is their high capacity for lipid accumulation, which are the carbon storage compounds accumulated during nitrogen-limiting conditions [6,7]. *Yarrowia* clade species vary in terms of the maximum growth temperature, ranging from 21 °C for *Y. alimentaria* to 37 °C for *Y. phangngaensis* [7].

Most of the species belonging to the *Yarrowia* clade originate from European, American, or Asian regions, being isolated from various natural habitats, which are also common for *Y. lipolytica*, such as dairy and meat products *(Y. alimentaria*, *Y. osloensis*, *Y. divulgata*, *Y. galli*, *Y. bubula*, and *Y. porcina*), mangrove or marine waters (*Y. keelungensis*, *Y. phangngaensis* and other strains of *Y. divulgata*), guts of insects or vertebrates (*Y. parophonii*, *Y. yakushimensis*, and *Y. hollandica*) and from fermented vegetable food (*Y. brassicae*). Numerous concerns about the safety of using *Yarrowia* clade are related to *Y. deformans* and some strains of *Y. galli*, which have been isolated from human tissues [8].

However, based on the aerobic nature and the rare ability to grow at 37 °C, *Y. lipolytica* has been classified as generally recognized as safe (GRAS) by the American Food and Drug Administration (FDA) for particular applications. It was also considered a “safe-to-use” organism based on an exhaustive literature assessment [9]. Furthermore, the unique biochemical characteristics of *Y. lipolytica* contributed to the irrefutable interest of scientists in that species and its large-scale use [10]. Nowadays, *Y. lipolytica* is commercially used to produce microbial proteins, single cell oil, organic acids, and polyhydroxy alcohols. In addition, several enzymes like RNases, phosphatases, esterases, lipases, and proteases are produced using *Y. lipolytica* cultures [11]. The high proteolytic and lipolytic activity contributed to the widespread distribution of *Y. lipolytica* in the commercial production of meat and dairy products [12].

The most studied proteolytic enzymes produced by *Y. lipolytica*, i.e., extracellular alkaline protease (Aep) and extracellular acid protease (Axp), are synthesized under starvation conditions at the end of the exponential growth phase in protein-rich media [13,14].

Studies by Enderlin and Ogrydziak (1994) classified Aep as a 32 kDa subtilisin-like protein encoded by the gene, secreted in the form of a 55 kDa glycosylated precursor [15]. To develop the mature form of this enzyme, a 15 amino acid long presequence had to be removed, starting with the cleavage of the dipeptides (X-Ala or X-Pro) followed by hydrolysis of the peptide bond between Lys and Arg [16]. Aep secretion by *Y. lipolytica* growing in a yeast extract peptone dextrose (YPD) medium (pH 6.8) reaches around 1–2 g/L [17]. Furthermore, as shown by Akpınar and colleagues (2011), skimmed milk-based medium is appropriate for Aep secretion, and no negative impact of additional carbon (glucose) or nitrogen (glutamine) sources was determined [18].

Early studies on the nature of the Axp protease conducted by Yamada and Ogrydziak (1983) indicated the formation of three mature forms of this enzyme, which differed in their molecular weights—28, 32, and 36 kDa [19]. However, Young and colleagues (1996) showed the formation of a 37 kDa enzyme, sometimes subject to partial hydrolysis. Consequently, Axp was finally classified as a pepsin-like protein, composed of 397 amino acids with a 44 amino acid long prepro-region. The enzyme, secreted as a precursor with a 42 kDa molecular mass, is converted into its mature form in acidic environments [20].

In contrast to *Y. lipolytica*, the information about the biotechnological applications of other *Yarrowia* clade members are very limited. For example, studies about erythritol, mannitol, and arabitol production, with pure or crude glycerol as well as with glucose or fructose as carbon sources, are still in the laboratory research phase [21,22]. Similarly, studies related to lipid biosynthesis in a medium based on switch grass hydrolysate using *Yarrowia* clade species were performed on a small scale [6].

According to our current knowledge, studies performed by Ozturk [23] are the only ones related to the proteolytic activity of *Yarrowia* species other than *Y. lipolytica*. These studies showed the ability of *Y. galli*, *Y. deformans*, and *Y. alimentaria* to gelatin liquefaction, their high proteolytic activity, and their ability to secrete proteases. Therefore, due to the presumably high application potential of the produced proteases, it is necessary to expand our knowledge about these enzymes and their properties.

The selection of a substrate in protease research is crucial for commercial application. The cost of the growth medium represents about 30–40% of the price of industrial enzymes [24]. The brewers’ spent grain (BSG), the major solid by-product in the brewing industry, due to the high protein content (19–30% *w*/*w*) and low price of approximately EURO 35 per ton, is a prospective substrate. Moreover, the average annual global production of BSG, estimated to be about 39 million tons, with 3.4 million tons produced in the European Union, forces the search for new valorization methods for this agro-industrial by-product [25].

Our research provides a comprehensive analysis of the alkaline proteolytic enzymes of *Yarrowia* clade species, including phylogenetic studies, synteny analysis, and protease production and application. Furthermore, in order to address the economic aspects of protease production, a protein-rich spent grains medium was selected. Finally, considering the potential applications of *Yarrowia* proteases, we established a link between the high degree of protein hydrolysis and antioxidant activity of the obtained hydrolysates.

## 2. Results and Discussion

### 2.1. Comparative Analysis of Yarrowia Alkaline Proteases from a Genomic Point of View

The search for alkaline proteases encoding genes in the genome of the different *Yarrowia* species showed the presence of 5 (*Y. phangngaensis*) to 21 (*Y. galli)* genes (Table 1). Five genes were also found in *C. hispaniensis*, which has a basal position in the clade. Due to the lack of access, genomes of *Y. brassicae* and *Y. parophonii* were not examined in terms of alkaline protease genes. Instead, three strains of *Y. lipolytica* were analyzed: the Polish strain A101, the German strain H222, and the laboratory strain E150 (CLIB 122) issued from a crossing between the French W29 and the American ATCC 18942 followed by many back-crosses [8]. The genome of E150 was the first fully sequenced genome and is still the reference for genome assembly and annotation of new *Y. lipolytica* genomes.

Based on sequence similarity, we defined 13 different groups of putative alkaline proteases as shown on the phylogenetic tree (Figure 1).

Most groups contained one *Y. lipolytica* representative except for three groups with two proteins with multiple counterparts that could not be separated based on sequence similarity. One of these groups corresponds to the *XPR2* group, i.e., the Aep encoded by YALI0F31889g, part of which (highlighted by a blue curve in Figure 1) gathered one protein per species except for *Y. phangngaensis* and *C. hispaniensis* for which no ortholog was found. The genetic distances between these genes are small, as shown by the short branches in the tree. The other proteins in the *XPR2* group exhibited very long branches representing proteins emanating from successive duplication events, which occurred in *Y. osloensis*, *Y. porcina*, *Y. galli,* and *Y. yakushimensis*. Synteny analysis of these genes showed different insertion sites suggesting that these duplications occurred independently (Appendix A).

One exception to this rule is the conservation of synteny in the upstream region of YAOS0S01e24608g and YAPO0S01e09780g genes, which are probably derived from the same event (Appendix A). In *Y. yakushimensis*, YAYA0S05e13234g, the syntenic copy of *XPR2*, was successively duplicated three times on scaffold YAYA0S6. These duplication events involved not only *XPR2* but also the homolog of YALI0F31845g (F31845 in Appendix A).

In contrast, there is extensive synteny conservation in the region of *XPR2* close homologs (part of the *XPR2* group with short branches) with some insertions of tRNA genes, transposable elements, and intervening genes (Figure 2). Nevertheless, we observed a rearrangement occurring one gene upstream of the *XPR2* homolog (upstream of YALI0F31867g) in *Y. bubula*. An additional rearrangement generated a loss of synteny one gene downstream of *XPR2* (downstream from YALI0F31911g). As synteny is conserved in *Y. bubula*, *Y. hollandica,* and *Y. alimentaria*, which are basal species of the genus *Yarrowia*, the most parsimonious explanation is that the rearrangement occurred in the other species (grey arrow in Figure 2), leading to the actual order of genes in *Y. lipolytica* and closely related species.

### 2.2. Expression Analysis of Y. lipolytica W29 Alkaline Proteases in Various Culture Conditions and Signal Peptide Prediction

To verify which of the 16 putative proteases may play an important role in protein hydrolysis by the analyzed strains, we took advantage of previously published RNA-seq data of *Y. lipolytica* W29 [26] and unpublished data performed on a derivative of E150 (CLIB 122) (genotype: E150, *ku70::URA3*, *LEU2*). Twelve putative protease encoding genes showed low or no expression (Table 2). The *XPR2* gene is highly expressed in minimal medium with glucose but poorly in oleic acid and tributyrin. YALI0A06435g and YALI0A08360g were moderately expressed on the three tested carbon sources in yeast nitrogen base (YNB) medium. Surprisingly, YALI0B16500g, a homolog of *Saccharomyces cerevisiae PRB1*- protease B, is highly expressed in YPD medium at high and low pH. Previous studies showed that *XPR2* is induced by pH above 6 and/or by the presence of peptone [27]. Thus, we checked the expression of all putative protease encoding genes in these conditions and, as expected, *XPR2* was expressed on the YPD medium, although moderately, probably due to the genotype of E150 (CLIB 122) derivative (*his-1*, *ura3-302*, *leu2-270*, *xpr2-322*, *ku70::URA3*, *LEU2*). In turn, low expression of *XPR2* was observed on YPD glucose at pH 4 (Table 2). The highest expression was observed for YALI0B16500g on both YPD and YPD at pH 4. Furthermore, the functional annotation of YALI0B16500g predicts it to be localized in the vacuole. Thus, we checked all proteases for the presence of a putative signal peptide. It appeared that all of the putative proteases, including YALI0B16500p, could potentially be secreted (Table 2).

### 2.3. Analysis of Expression of XPR2Alkaline Protease in the BSG Medium by Y. lipolytica

Expression analysis of all the genes encoding putative proteases in the BSG-based medium was performed. Due to the large number of genes to be tested, we analyzed the expression only in the *Y. lipolytica* W29 strain. Out of the 16 genes encoding putative proteases, 7 were expressed under the analyzed conditions. The highest expression was observed for *XPR2,* followed by the YALI0B16500 gene (Figure 3). These data are, however, in contrast with the proteomic studies, in which the analyzed bands on SDS-PAGE were covered mostly with Aep protein (details see Section 2.7). These results were further verified by overexpression of the YALI0B16500 gene in the PO1d strain (deleted for the *XPR2* gene), where no protease activity was detected on skimmed milk agar medium at pH 6.8.

### 2.4. Investigation of the Proteolytic Activity of Yarrowia Clade Species

The proteolytic activity of the *Yarrowia* clade was analyzed using skimmed milk agar plates. Results showed huge diversity of the clade members in terms of their abilities to secrete proteases (Table 3).

*Y. phangngaensis*, *Y. deformans,* and *C. hispaniensis* showed no proteolytic activity, indicated by the lack of proteolysis zones on the skimmed milk agar (Table 3). The highest proteolytic activity was observed for *Y. yakushimensis* (15.05 ± 0.52), followed by *Y. lipolytica* (14.80 ± 0.40), *Y. keelungensis* (14.02 ± 0.04), and *Y. alimentaria* (13.51 ± 1.01) (Figure 4).

The proteolytic activity of fungal, yeast, and bacterial strains derived from an artisanal raw ewe’s milk cheese was investigated by Ozturkoglu-Budak and colleagues [28]. Among the five tested yeast species, only *Y. lipolytica* AF_335977 and *Debaryomyces hansenii* AF_210327 were able to secrete proteases. However, the proteolytic activity of AF_335977 was described as medium. At the same time, it was the best result among tested yeast species. In the studies by Gardini and colleagues [29], several yeast species isolated at different stages of Pecorino Crotonese production and ripening were characterized by their technological properties. However, only five strains, identified as *Y. lipolytica* and *D. hansenii,* were able to hydrolyze skimmed milk proteins. 

Furthermore, *Y. lipolytica* and numerous other yeast species isolated from starter-free raw cow’s milk cheeses were investigated for extracellular proteolytic activity. The highest activity of proteolytic enzymes was observed for *Y. lipolytica* and *K. lactis*, while the highest caseinolytic activity was presented by *Y. lipolytica* and *P. guilliermondii* [30]. Similarly, the proteolytic activity of yeast species isolated at different stages during the processing of pastirma (the most popular dry-cured meat product in Turkey) was investigated [23]. The yeast isolates, classified as *Candida zeylanoides* (the predominant species), *C. deformans*, *C. galli* as well as *C. alimentaria* (currently classified as *Yarrowia* genus), showed clear zones surrounding the colonies. Our results clearly confirm the potential of some *Yarrowia* species to secrete high amounts of proteases into the growth medium.

### 2.5. Characterization of the Proteolytic Activity of Yarrowia Clade Growing on BSG

The *Yarrowia* clade members differed significantly in terms of their proteolytic abilities. *Y. phangngaensis*, *Y. deformans,* and *C. hispaniensis* showed no proteolytic activity under the analyzed conditions (Figure 5). In *Y. lipolytica* W29, *XPR2* is the highest expressed gene in the BSG medium (Figure 3). The absence of proteolytic activity in *Y. phangngaensis* and *C. hispaniensis* could be explained by the loss of the *XPR2* homolog (Figure 1), which is, however, not the case in *Y. deformans.* The latter species was analyzed for the expression of *XPR2* ortholog (YADE0S25e00298g) in BSG medium. However, no transcript was detected.

The proteolytic activity of *Y. hollandica*, *Y. porcina*, *Y. yakushimensis*, *Y. bubula*, *Y. osloensis*, *Y. divulgata*, and *Y. brassicae* did not exceed 0.5 U/mL (Figure 5). The highest proteolytic activity was detected for *Y. lipolytica* W29 with 2.47 U/mL/min (Figure 5). Other *Yarrowia* species showed strong proteolytic activities: *Y. galli* (1.96 U/mL/min), *Y. alimentaria* (1.38 U/mL/min), *Y. keelungensis* (1.07 U/mL/min), and *Y. parophonii* (0.80 U/mL/min). These results allow us to consider the above-mentioned strains as interesting producers of proteolytic enzymes with potential industrial applications in the future.

In the cultures of *Y. lipolytica*, *Y. alimentaria,* and *Y. parophonii*, the proteolytic activity remained stable throughout the process, indicating continuous enzyme secretion and protein hydrolysis. In turn, in cultures of *Y. galli* and *Y. keelungensis,* a significant decrease in protease activity after reaching the maximum point was observed. It is possible that *Y. galli* and *Y. keelungensis* secreted a significant amount of enzymes at the beginning of the culture, allowing rapid protein hydrolysis and providing sufficient nutrients to decrease protease secretion.

The proteolytic activity of *Y. lipolytica* W29 and *Y. lipolytica* IMUFRJ 50862, isolated from an estuary near Rio de Janeiro, were analyzed in the research of Braga and colleagues [31], who investigated the effect of lipase secretion on the γ-decalactone production. In a medium optimized for lipase production, the proteolytic activity reached 0.0034 U/mL/min and 0.00151 U/mL/min for W29 and IMUFRJ 50862, respectively. The difference in results obtained in our work, caused by the medium composition, clearly indicates the potential for using BSG as a substrate.

In the present work, high proteolytic activity was associated with significant hydrolysis of BSG protein. Furthermore, it caused the accumulation of hydrolysis products, expressed as free α-amino groups. In the culture of *Y. lipolytica*, *Y. galli*, *Y. alimentaria*, and *Y. keelungensis*, characterized already as leading proteolytic strains, the concentration of α-amino groups reached 4367μg/mL, 4245 μg/mL, 3939 μg/mL, and 3885 μg/mL, respectively (Figure 6).

Interestingly, in our previous work using proteolytic bacteria, the concentration of free α-amino groups in the same BSG-based medium never exceeded 2600 μg/mL [32]. This result clearly shows the benefits of using *Yarrowia* clade yeast when applying BSG.

Eight strains of *Y. lipolytica*, isolated from goat/water buffalo mozzarella cheese, were previously studied by Suzzi and colleagues [33] for their ability to breakdown milk proteins. *Y. lipolytica* isolates showed weak proteolytic activity (expressed as leucine concentration [mg] per 100 mL of milk) after 8 days of the process. However, between the 8th to the 14th day, a huge increase in proteolytic activity was observed. After 14 days, the concentration of leucine reached 300–600 mg/100 mL, while in our research, the concentration of free α-amino groups in the BSG medium reached a maximum of 350–450 mg/100 mL for the most active strains (Figure 6). It is worth pointing out that the time of culture in our research was only 5 days. This result further confirms the proteolytic abilities of *Yarrowia* clade members. For six out of fifteen tested species, the degree of BSG protein hydrolysis (DH%) exceeded 50% (Figure 7).

The best results were observed for *Y. lipolytica* with 72.78% hydrolyzed proteins. Therefore, we conclude that the hydrolysis performed by *Yarrowia* species enabled a major protein breakdown. Furthermore, in similar studies with *Rhizopus* sp., DH% of the BSG protein reached 59.0 ± 5.4% [34]. This further confirms that BSG proteins are strong inducers of the *Yarrowia* clade proteolytic enzymes and, thus, that it is an appropriate substrate for optimization studies. Moreover, given that filamentous fungi are common producers of hydrolytic enzymes, the results obtained by *Y. lipolytica*, *Y. galli*, *Y. alimentaria,* and *Y. keelungensis* are more than satisfactory.

The protein hydrolysis in the cultures of *Yarrowia* clade members was associated with environmental alkalization (Figure 8). As a result, the final pH fluctuated between 8.5 and 9. Only in the *Y. phangngaensis*, *Y. deformans,* and *C. hispaniensis* culture, where a lack of proteolytic activity and no hydrolysis of the BSG proteins were observed, did the pH not exceed 8.

Medium alkalization in the cultures of *Y. lipolytica* confirmed the previous studies in which *K. lactis*, *D. hansenii*, *Pichia guilliermondii*, *Pichia fermentans*, *S. cerevisiae*, *Candida* spp., and *Rhodotorula* spp., as well as 21 strains of Y. lipolytica, were analyzed for their properties influencing cheese production [30]. As a result, 16/21 and 10/21 of *Y. lipolytica* strains showed alkalizing properties on citrate agar and litmus milk, respectively.

To summarize the current information, species such as *Y. lipolytica*, *Y. alimentaria*, *Y. galli,* and *Y. keelungensis* are the most efficient and universal (in terms of medium composition) proteolytic enzyme producers.

### 2.6. Profile of Yarrowia Clade Proteolytic Enzymes Secreted in BSG-Based Medium

The gelatin zymography allowed for the detection of several proteolytic enzymes produced extracellularly during the growth of *Yarrowia* clade species in BSG medium (Appendix A). Molecular masses of proteolysis bands observed on zymography gel are summarized in Table 4. It is important to underline that some proteolysis bands represent protease complexes formed as a result of interactions between enzyme monomers. This can be confirmed in the *Y. lipolytica* example. The molecular weight of *Y. lipolytica* proteases is 40–90 kDa. Therefore, producing an enzyme with a molecular weight of 110 or 130 kDa is impossible. Nevertheless, based on obtained results, it is possible to define the enzymes/enzymes complex involved in BSG protein hydrolysis.

As shown, *Y. lipolytica* and *Y. alimentaria* were the most active species in proteolytic activity and the most efficient in BSG protein hydrolysis. The protease profile of *Y. lipolytica* and *Y. alimentaria* was represented by two main proteolytic bands at 110 and 130 kDa (Table 4). However, a proteolytic band at 130 kDa was also detected in *Y. brassicae* and *Y. divulgata* culture, where low proteolytic activity and DH% < 30 were estimated. Therefore, we suggest a key role of the 110 kDa protein complex in BSG protein hydrolysis.

Furthermore, strong proteolytic activity in BSG medium as well as significant DH% of BSG protein, was demonstrated for *Y. galli*. The profile of proteolytic enzymes of *Y. galli* was represented by two bands with a molecular mass of 37–39 kDa and 40 kDa (Table 4). Nevertheless, the production of 40 kDa protease of low proteolytic activity (<0.5 U/mL) was determined in the *Y. osloensis* culture medium. Following this, protease of 37–39 kDa seems significant for BSG protein hydrolysis. It is important to underline that the same enzyme profile as *Y. galli* was observed for *Y. parophonii*, in which proteolytic activity (0.80 U/mL/min) was almost twice as low as *Y. galli* (1.96 U/mL/min). An explanation for this is the difference in the expression level of the 37–39 kDa enzyme.

Finally, a protease of molecular mass 30 kDa, secreted in *Y. keelungensis* (Table 4), is a crucial factor affecting BSG protein hydrolysis as indicated by high proteolytic activity and significant DH%. Production of proteolytic enzymes of 30 kDa was also observed in the *Y. porcina* culture medium. However, based on the results of the proteomic analysis presented in Section 2.7, enzymes of *Y. keelungensis* and *Y. porcina* are, with a high probability, different compounds of similar molecular weight.

Production of proteolytic enzymes in *Y. lipolytica* was already reported. An alkaline extracellular protease of 35 kDa was produced by *Y. lipolytica*YITun15 [35]. This enzyme showed significant activity in a wide spectrum of factors tested. A *Y. lipolytica* coconut copra isolate produced protease of 61.5 kDa. The fish flower was applied as a substrate and enzyme production inductor [36]. In other studies, a 28 kDa protease was secreted by an auxotrophic recombinant strain of *Y. lipolytica* Po1d (JMY 1852). The enzyme was produced as an undesirable factor affecting human interferon alpha 2b production [37]. Another example involved the production of the intercellular protease of 97 kDa by *Y. lipolytica* strain IPN-MI-1 of the yogurt origin. The enzyme showed stable activity in the pH range of 5.0–9.0, and 37 °C was the optimal temperature [38].

In order to conduct proteomic analysis of *Yarrowia* clade enzymes, protein electrophoreses were performed. Unfortunately, despite a significant increase in protease concentration obtained using acetone precipitation, some of the proteases detected by gelatin zymography did not appear on the electrophoretic gel. Therefore, only the samples presented in Appendix A were subjected to proteomic investigation.

### 2.7. Protein Sequence Coverage of Yarrowia Clade Proteases to Other Fungal Enzymes

Proteomic analysis of the enzymes expressed in the BSG medium culture of *Y*. *lipolytica*, *Y. keelungensis,* and *Y. alimentaria* showed high protein sequence similarity to Aep of *Y. lipolytica* E150 (CLIB122) encoded by *XPR2* (Appendix A). The amino acid sequence of the *Y*. *lipolytica* enzyme complex of 130 kDa and 110 kDa was 40% and 44% aligned to the sequence of Aep, respectively. Moreover, the protein sequence coverage of the *Y. alimentaria* enzyme complex (130 kDa) and *Y. keelungensis* enzyme (30 kDa) to Aep reached 42% and 35%, respectively.

In the analysis of these results, two factors were important. First, we used public databases of fungal proteins to analyze the proteomic data, but the proteomes of *Yarrowia* species are not available, except those of *Y. lipolytica* strains. Despite this, the results of the proteomic analysis confirmed close affinity between *Yarrowia* species. Moreover, matched peptide sequences were found in the position range of 158–454, corresponding to the proper chain of Aep, which is converted into the active form of the enzyme upon signal peptide and propeptide cleavage.

The proteolytic enzyme of *Y. osloensis* (65 kDa) and *Y. parophonii* (65 kDa) were, to a great extent, 32% and 26%, respectively, aligned to alkaline extracellular protease of *Yarrowia* sp. E02 and *Yarrowia* sp. B02, while sequence coverage of *Y. porcina* (30 kDa) protease to the Carboxypeptidase of *Y. lipolytica* YB392 coded by B0I71DRAFT_162965 gene reached 25%. The full set of *Yarrowia* clade enzymes proteomic analysis results is available in the Appendix A (Appendix A).

### 2.8. Antioxidant Activity of BSG Protein Hydrolysates

Three methods were applied in order to estimate the antioxidant activity of BSG protein hydrolysates obtained in the cultures of *Yarrowia* clade species. Significant antioxidant activity was detected using the ABTS method. The highest activity, reaching 2606.13 μM TEAC/g, was detected in the *Y. divulgata* culture medium (Figure 9). This result was followed by the activity of *Y. galli*, *Y. keelungensis,* and *Y. lipolytica* samples, reaching 1771.87 μM TEAC/g, 1730.73 μM TEAC/g, and 1722.63 μM TEAC/g, respectively. It was more than twice higher than the activity of the control sample (603.73 μM TEAC/g). According to our previous research, the antioxidant activity of the control samples is caused by the partial hydrolysis of BSG protein occurring during media sterilization [32]. High antioxidant activity of post-culture media of *Y. galli*, *Y. keelungensis*, and *Y. lipolytica* strains resulted mainly from the efficient hydrolysis of spent grain protein (DH% > 60%), which is not the case of *Y. divulgata*, where DH% did not exceed 30%.

In studies where protein hydrolysis is carried out directly in microbial cultures, the antioxidant activity of hydrolysates may be affected by the products of microbial metabolism. BSG is a waste material containing, in addition to protein, cellulose (17–25%), non-cellulosic carbohydrates (25–35%), lignin (8–28%), lipids (10%), and ash (5%) [39]. Therefore, due to the hydrolysis, other compounds, such as phenolics, may be exposed or released to the culture medium affecting the antioxidant activity of hydrolysate [40]. *Y. divulgata* is the object of extensive studies concerning erythritol production [22,41]. According to den Hartog et al. (2010), erythritol was recognized to possess strong antioxidant activity [42]. Therefore, simultaneous production of erythritol as a process accompanying BSG protein hydrolysis should be considered possible.

In our previous work, the antioxidant activity of BSG protein hydrolysates, obtained using the ABTS method directly in proteolytic bacteria cultures, ranged from 345.18 to 1621.31 μM TEAC/g peptides [32]. In this work, most of the analyzed samples showed a total activity of about 1500 μM TEAC/g. Therefore, using some *Yarrowia* clade yeast proteolytic enzymes allowed us to obtain hydrolysates with higher antioxidant activity compared to bacterial enzymes. It is likely related to DH%, reaching higher values in the yeast cultures, showing that high antioxidant activity is strongly associated with a high degree of protein breakdown.

The other two methods did not demonstrate the significant antioxidant activity of hydrolysates compared to the control. The best result obtained using the DPPH method is 144.90 μM TEAC/g in the culture medium of *Y. hollandica*, i.e., almost 10 times lower than the activity detected using the ABTS method (Figure 9). The antioxidant activity of *Yarrowia* clade culture samples did not reach the control activity (147.80 μM TEAC/g). However, these results align with the work of Wołosiak and colleagues [43], in which the limitation in the solubility of peptides and amino acids in the system of DPPH assay (free radical resuspended in ethanol solution) is underlined. In addition, numerous problems related to the formation of iridescence in the presence of amide compounds were described. Therefore, the authors recommended the ABTS method as appropriate for the research on the samples containing significant quantities of peptides. On the other hand, due to large differences in the DPPH and ABTS systems, when possible, it is still advisable to apply both methods.

The antioxidant activity of BSG protein hydrolysates was evaluated using a ferric reducing antioxidant power (FRAP) assay in the work of Vieira and colleagues [44]. Activities of hydrolysates reached 0.062 mg TE/mg dw (247.71 84 μM TEAC/g), 0.098 mg TE/mg dw (391.55 μM TEAC/g), and 0.101 mg TE/mg dw (403.53 μM TEAC/g) when using brewery spent yeast extract, neutrase, and alcalase, respectively, for hydrolysis purposes. According to Vieira and colleagues, FRAP of protein hydrolysates depends not only on the molecular size of peptides but on the presence of specific peptide sequences and the amino acid composition of the hydrolysis products.

Antioxidant activities measured in *Yarrowia* clade cultures ranged between 140–240 μM TEAC/g using the FRAP method, with the highest result obtained for the hydrolysate of *Y. divulgata* (233.13 μM TEAC/g) (Figure 9). In addition, the obtained activities were not significantly different from the control sample (181.46 μM TEAC/g). Presumably, due to their chemical properties, the peptides in our samples did not precipitate in the reduction reaction of Fe^3+^ into Fe^2+^ as electron donors, which is the basis of the FRAP method.

Nowadays, antioxidant peptides are increasingly applied in the production of nutraceuticals and functional food ingredients [45], pharmaceuticals [46], and in the field of dermatology [47]. As an aging population, we search for ingredients of increased potency. Therefore, peptides of the activity that result in diminished lines and wrinkles, smoother skin texture, and reduced redness and skin discoloration are highly sought after in the skincare industry [47]. Moreover, the unstable and reactive nature of free radicals results in cell damage leading to diseases such as hypertension, cardiovascular disease, cancer, diabetes mellitus, and neurodegenerative and inflammatory diseases. Therefore, antioxidant peptides protect the body by scavenging free radicals and inhibiting lipid peroxidation reactions, thus preventing oxidative damage [46]. In addition, antioxidant peptides are also applied in food systems to preserve food products against lipid oxidation leading to the development of undesirable off-flavors, odors, and potentially toxic reaction products [45]. Therefore, we believe that the peptides obtained as a result of BSG protein hydrolysis can be used in the food, pharmaceutical, and medical industries.

## 3. Materials and Methods

### 3.1. Chemicals, Microorganisms, and Materials

Ninhydrin (analytical grade), Trizma^®^ base (≥99.9%), Trichloroacetic acid (TCA, ≥99.0%), Triton X-100 (analytical grade), Sodium Dodecyl Sulfate (SDS; ≥99%), Bromophenol Blue (analytical grade), 2,2′-azino-di-(3-ethylbenzthiazoline sulfonic acid) (ABTS, ≥98%)), potassium peroxodisulfate (≥99%), 2,2-diphenyl-1-picrylhydrazyl (DPPH), 6-hydroxy-2,5,7,8-tetramethylchroman-2-carboxylic acid (Trolox, ≥97%), iron(III) chloride hexahydrate (FeCl_3_ × 6H_2_O, ≥97%),2,4,6-Tris(2-pyridyl)-S-triazine (TPTZ, ≥98%), Coomassie Brilliant Blue R 250 were purchased from Sigma (Sigma-Aldrich, Saint-Louis, MO, USA) and 4×Laemmli Sample Buffer was purchased from Bio-Rad (Bio-Rad Laboratories, Hercules, CA, USA). All other chemicals used in the following studies were of analytical grade.

*Yarrowia* clade species: *Y. lipolytica* W29, *Y. hollandica* CBS 4855, *Y. galli* CBS 9722, *Y. phangngaensis* CBS 10407, *Y. deformans* CBS 2071, *Y. keelungensis* CBS 11062, *Y. porcina* CBS 12935, *Y. yakushimensis* CBS 10254, *Candida hispaniensis* CBS 9996, *Y. bubula* CBS 12934, *Y. osloensis* CBS 10146, *Y. alimentaria* CBS 10151, *Y. divulgata* CBS 11013, *Y. parophonii* CBS 12427, and *Y. brassicae* CBS 15225 were purchased from the Centraalbureau voor Schimmel cultures (CBS, Utrecht, The Netherlands), stored in YPD(agar (15 g/L), bacteriological peptone (20 g/L), glucose (20 g/L), yeast extract (10 g/L)) slants and cultured in YPD broth at 25 °C, 180 rpm for 24 h before use.

BSG, used as a nutrient source and factor inducing the proteolytic enzyme secretion, was purchased from the Polish brewery Browar Stu Mostów (Wroclaw, Poland), where it was collected after wort production. Initial wort was obtained from the malt composed of 30% barley and 70% wheat grains and therefore contained various proteins. Initially, in order to protect the raw material from the development of contaminating microflora, raw BSG was dried at 50 °C for approximately 48 h. During this process, the material was periodically mixed and controlled in terms of humidity.

### 3.2. Bioinformatic Analysis of Yarrowia Clade Proteolytic Enzymes

Protease sequences were extracted from the *Yarrowia* genomes previously published by Červenák and colleagues [48] by sequence homology to alkaline extracellular protease *XPR2* (YALIA101S03e00364g) from *Y. lipolytica* A101. The blastp threshold for E-value was set to 1 × 10^−20^. Protein sequences were aligned using Clustal Omega [49] in the multi-platform graphical user interface Seaview version 4.6.2 [50]. Gblocks [51] was used to select blocks of conserved sites useful for phylogenetic analysis. Trees were reconstructed by maximum likelihood using PhyML 3.0 [52] with an LG evolutionary model. Robustness of the trees was assessed by the approximate likelihood ratio test approach (aLRT). Synteny analyses were done manually using Artemis genome browser [53].

### 3.3. Analysis of Alkaline Proteases Expression in Y. lipolytica Grown in Various Culture Conditions and Prediction of Putative Signal Peptide

Previously published data [26] were used to estimate the expression level of each *Y. lipolytica* W29 alkaline protease on yeast nitrogen base (YNB) with oleic acid, glucose, and tributyrin. Additional unpublished data from cDNA sequence analysis (Neuvéglise and colleagues) were used for analyzing the expression of the same genes in YPD and YPD at pH 4.0. Signal peptide predictions were performed with SignalP-5.0 [54] and DeepLoc-1.0 [55].

### 3.4. Analysis of Alkaline Proteases Expression in Y. lipolytica Grown on Spent Grains

The analysis of gene expression encoding verified and putative alkaline protease in *Y. lipolytica* was performed in the yeast cells growing on BSG according to the methodology described in Section 3.5. After 72 h of culture, the biomass was filtered through Whatman filters and collected into 50 mL Falcon tubes. The biomass was divided into 2 mL samples, centrifuged at 15,000 rpm for 1 min, washed 3 times with sterile distilled water, and frozen in liquid nitrogen. RNA was extracted from three biological replicates using RNeasy Micro Kit (Qiagen; Santa Clarita, CA, USA) following the manufacturer’s instructions. The obtained RNA was DNase digested (Thermo Fisher, Waltham, MI, USA) for reverse transcription reaction using Maxima First Strand cDNA Synthesis Kit (Thermo Fisher Scientific, Inc., Waltham, MA, USA). The obtained cDNA was a template for PCR reaction using Phire Green Hot Start II PCR Master Mix (Thermo Fisher Scientific, Inc., Waltham, MA, USA) and primers listed in Table 5. Similarly, expression of *Y. deformans* YADE0S25e00298g gene was analyzed following the above-described methodology. Actin gene (ACT1, YALI0D08272g) was chosen as a housekeeping gene in this analysis.

### 3.5. Cultivation of Yarrowia Clade Species in the BSG Medium

The analyzed species belonging to the *Yarrowia* clade were cultured in a minimal medium supplemented with BSG as carbon and nitrogen sources to stimulate proteolytic enzyme secretion. Yeast cultures were performed in 250 mL Erlenmeyer flasks containing 50 mL of the medium composed of BSG (50 g/L), MgSO_4_ (1 g/L), KH_2_PO_4_ (0.1 g/L), K_2_HPO_4_ (0.13 g/L), CaCl_2_ (0.5 g/L), and FeSO_4_ × H_2_O (0.01 g/L). After sterilization, the pH of the medium was adjusted to 7.0 by adding sterile 5 M NaOH (addition of approx. 320 μL per flask). Before inoculating the main culture, yeast was grown in YPD medium for 24 h under180 rpm agitation at 25 °C, centrifuged for 10 min at 7500 rpm, and washed with distilled water. The optical density was standardized to reach OD_600_ = 0.25 in the BSG medium. Non-inoculated medium was cultured under the same conditions and used as a control. Each culture was cultivated for 168 h at 25 °C on a rotary agitation (180 rpm). One milliliter samples were taken every 48 h and analyzed for proteolytic activity, and the concentration of free α-amino groups was estimated. As pH is an important factor affecting the type of proteases produced by *Yarrowia* clade species, appropriate pH measurements were performed.

### 3.6. Determination of Hydrolysis Efficiency

In order to define involvement of proteases secreted by *Yarrowia* clade species on the availability and solubilization of BSG proteins, %DH was estimated. For this purpose, the concentration of free α-amino groups was determined using ninhydrin method [56]. The maximum concentration of α-amino groups detected for each culture was compared to the data obtained for the protein hydrolysis using hydrochloric acid (total hydrolysis), performed as described previously [32]. The %DH was calculated according to the following equation:(1)$$\%DH=\left(BH\times 100\right)/TH$$
where TH (μg/mL) and BH (μg/mL) are the average concentration of α-amino groups obtained, respectively, as a result of the total hydrolysis (TH) or in the yeast culture (BH) calculated per 1 g of BSG sample.

### 3.7. Analysis of the Proteolytic Activity

In order to evaluate the proteolytic activity of the yeast species, the method of Anson was applied [57]. Shortly, proteolytic activity was estimated by mixing 0.45 mL of casein (1.1% in 50 mM Tris-HCl buffer, pH 7.5) with 0.5 mL of 50 mM Tris-HCl buffer (pH 7.5) and 0.05 mL of the analyzed sample. Reaction mixtures were incubated at 55 °C for 10 min, followed by adding 0.5 mL 8% (*w*/*v*) TCA to terminate the reaction. For each sample, appropriate negative controls were performed (no enzyme addition). As a positive control, protease from *Bacillus licheniformis* (Sigma-Aldrich, Germany) was applied. Finally, samples were centrifuged at 12,000 rpm for 15 min, and the absorbance of supernatants was determined at 280 nm. Tyrosine was used as a standard in the concentration range of 0–2 mM/L. One unit of enzyme activity was defined as the amount of enzyme required to release 1 μM of tyrosine per 1 mL of culture medium within 1 min under the experimental conditions.

The proteolytic activity of the *Yarrowia* clade species was also detected using skimmed milk agar plates. For this purpose, the standardized biomass solutions (OD_600_ = 1.0) were spotted (5 μL) on the agar plates containing 2% (*w*/*v*) of agar and 10% of sterile skimmed milk (0.5%) (*w*/*v*). Inoculated plates were air dried under sterile conditions followed by 24 h incubation at 25 °C. Finally, the proteolysis zones were measured. As a control, non-inoculated medium was spotted on the agar plates and incubated under the same conditions. Both types of proteolytic activity measurements were performed in triplicate.

### 3.8. Estimation of Proteolytic Enzyme Profiles

#### 3.8.1. Determination of the Profile and Molecular Mass of *Yarrowia* Clade Proteases Using Gelatin Zymography and Electrophoresis

Gelatin zymography was carried out according to the methodology described previously [32]. The samples, collected after 72 h of *Yarrowia* clade cultivation in BSG medium, were filtered through 0.45 μm membrane and concentrated using Amicon Centrifugal Filters in order to apply approximately 5μg of proteins on a gel well. Finally, samples were mixed 1:1 with a Sample buffer (62.5 mM Tris–HCl (pH 6.8), 0.01% bromophenol blue, 2% SDS, and 25% glycerol), cooled down, and subjected to SDS-PAGE with a 10% gel containing 0.1% gelatin. Thermo Scientific™ PageRuler™ Prestained Protein Ladder (Thermo Fisher Scientific, Cleveland, OH, USA), 10 to 180 kDa was used as a molecular weight standard. The non-inoculated BSG medium was used as a control.

In order to conduct proteomic analysis of secreted proteases, SDS-PAGE electrophoresis was performed. Samples were collected at 72 h, and proteins were precipitated using acetone. Each sample, 400 μL, was cooled down on ice and mixed with 1600 μL of acetone, previously cooled to −80 °C. Mixtures were shaken vigorously and incubated at −80 °C for 1 h. Samples were centrifuged at 13,000 rpm for 15 min. Supernatants were discarded, and final protein precipitates were evaporated using vacuum evaporator to remove residual acetone. Protein precipitates were dissolved in the 50 μL of 50 mM Tris-HCl buffer, pH 7.4, in order to obtain 8 times concentrated samples. For strains *Y. divulgata*, *Y. hollandica*, *Y. bubula*, *Y. brassicae*, and *Y. yakushimensis,* the described procedure was not enough to obtain positive result of electrophoresis. The process was thus repeated with 1 mL of the medium concentrated 20-fold.

In order to run the SDS-PAGE, 8- and 20-times concentrated samples were mixed with 4×Laemmli Sample Buffer and loaded on 10% gel. Gels were run at 150 V followed by staining with Coomassie Brilliant Blue R250 (40% methanol, 10% acetic acid, 0.1% Coomassie blue) for 1 h and destained overnight (40% methanol, 10% acetic acid). Thermo Scientific™ PageRuler™ Prestained Protein Ladder, 10 to 180 kDa, was applied as a molecular weight standard. Uninoculated BSG medium was used as a control.

#### 3.8.2. Comparative Proteomic Analysis of the *Yarrowia* Clade Proteolytic Enzymes

Protein bands cut out from the SDS-PAGE gels were analyzed in the Laboratory of Mass Spectrometry, Institute of Biochemistry and Biophysics of Polish Academy of Sciences (Warsaw, Poland). Gel pieces were dried with ACN and reduced (treated) with 10 mM DTT in 100 mM NH_4_HCO_3_ for 30 min at 57 °C. Cysteines were then alkylated with 0.5 M iodoacetamide in 100 mM NH_4_HCO_3_ (45 min in dark at room temperature). Proteins were digested overnight at 37 °C with 10 ng/μL of trypsin in 25 mM NH_4_HCO_3_ (Promega). Afterward, samples were concentrated and desalted on an RP-C18 pre-column (Waters), and further peptide separation was achieved on a nano-Ultra Performance Liquid Chromatography (UPLC) RP-C18 column (Waters, BEH130 C18 column, 75 μm i.d., 250 mm long). NanoACQUITY UPLC system, using a 45 min linear acetonitrile gradient, was applied. Column outlet was directly coupled to the Electrospray ionization (ESI) ion source of the Orbitrap Elite mass spectrometer (Thermo), working in the regime of data-dependent MS to MS/MS switch with HCD peptide fragmentation. The raw data files were pre-processed with Mascot Distiller software (version 2.7, MatrixScience). The obtained peptide masses and fragmentation spectra were compared to libraries using the National Center Biotechnology Information (NCBI) non-redundant database nr 20210206 (349124773 sequences; 126215745164 residues), with “other Fungi” filter (16225947 sequences) using the Mascot search engine (Mascot Server v. 2.7, MatrixScience). The following search parameters were applied: enzyme specificity was set to trypsin, fixed modifications: Carbamidomethyl (C), variable modification: Oxidation (M), and one missed cleavage allowed. Finally, offline mass recalibration of peptide and fragment masses in individual LC-MS/MS runs was performed to reduce mass errors, with resulting parent mass window of around 5 ppm and fragment mass window of around 0.01 Da. Decoy Mascot option was activated, but the FDR computations were unsuccessful due to vastness of database and low number of spectra originating from samples of limited complexity. Therefore Mascot-provided score threshold of 45 was used for peptide/protein identification.

### 3.9. Estimation of Antioxidant Activity of BSG Protein Hydrolysates

The antioxidant activity of the hydrolysates was investigated using two methods based on the depletion of the radical cation, ABTS and DPPH assay, and the FRAP method involving reduction of colorless, oxidized Fe^3+^ form of iron into a blue-colored Fe^2+^ form. All determinations were performed according to Ciurko et al. [58]. Despite a similar mechanism of action, there are significant differences between ABTS and 2,2-diphenyl-1-picrylhydrazyl (DPPH) method, mostly related to the solubility of free radicals. In this research, ABTS radical was dissolved in water. Therefore, it was appropriate to estimate activity of water-soluble antioxidants, while DPPH radical, dissolved in ethanol solution, is more suitable to estimate activity of hydrophobic compounds. According to the work of Wołosiak and colleagues [43], deactivation of DPPH radical occurs predominantly via single electron transfer (SET) and uncommonly by hydrogen atom transfer (HAT), which has a dominant in the neutralization of ABTS radical. Differences in the described inactivation system cause recommendations for using both methods to determine the antioxidant activity of tested compounds. The antioxidant activity was estimated in samples after 120 h of the culture when DH% reached maximum value. Some modifications of DPPH assay were made: 10 μL samples were mixed with 190 μL of 0.1 mM DPPH radical solution in Eppendorf tubes (1.5 mL). Samples were incubated for 30 min in dark conditions, followed by centrifugation at 12,000 rpm for 6 min. After centrifugation, 100 μL of each sample was transferred to the microplate wells, and the absorbance at 517 nm was measured using Tecan instrument (Tecan Group Ltd., Männedorf, Switzerland). As a control, uninoculated medium incubated in the same conditions was used. Trolox (a synthetic analog of alpha-tocopherol) was used to estimate Trolox equivalent antioxidant capacity. Standard curves were prepared for 0–2000 μM, 0–1000 μM, and 0–400 μM for ABTS, DPPH, and FRAP methods, respectively. The obtained results were calculated per gram of protein hydrolysis products. For this purpose, peptide concentration in every sample was analyzed according to the Lowry method [59], with some modifications. A sample with a value of 10 μL was mixed with 50 μL AB solution and incubated for 10 min at room temperature. AB solution was prepared by mixing reagent A, composed of 0.1 M NaOH and 0.2 M Na_2_CO_3_, with reagent B containing 1% sodium citrate in a ratio of 50:1. After 10 min, 5 μL of Folin–Ciocalteau reagent was added to the mixture and incubated for additional 30 min at room temperature. Finally, the absorbance of each sample was measured at 750 nm using Tecan microplate reader. Peptide concentration was measured using peptide digest standard (Thermo Fisher Scientific, Cleveland, OH, USA) applied in a concentration range of 0–1000 μg/mL. The results were expressed as mean ± SD. Statistical significance was determined using Student’s *t*-test. The significance level was set at *p* < 0.05.

## 4. Conclusions

The major conclusion of this work is that species of the *Yarrowia* clade showed an important diversity of alkaline proteases with different secretion abilities. In addition, the obtained data showed that the BSG could be used as a substrate for *Yarrowia* clade yeasts to obtain protein hydrolysates with antioxidant activity. Results presented in our work clearly confirm the potential of *Yarrowia* clade species to secrete high amounts of proteases into the growth medium.

The presented research is the basis for scientists who would like to develop knowledge about the proteolytic enzymes of the *Yarrowia* clade leading to their widespread commercial use. Based on the data presented here, further research can be performed on optimizing protease or bioreactor scale production. Another direction of further research may be the development of protease purification technology. In this study, proteolytic enzymes were used to produce bioactive peptides as a consequence of BSG protein breakdown, and hydrolysates with significant antioxidant activity were obtained. We believe that the presented research can contribute to the development of a system of biopeptide production, applied as a nutraceutical and functional food ingredient or as pharmaceutical and active cosmetic ingredients for the skin care industry.

## Figures and Tables

**Figure 1 ijms-24-06514-f001:**
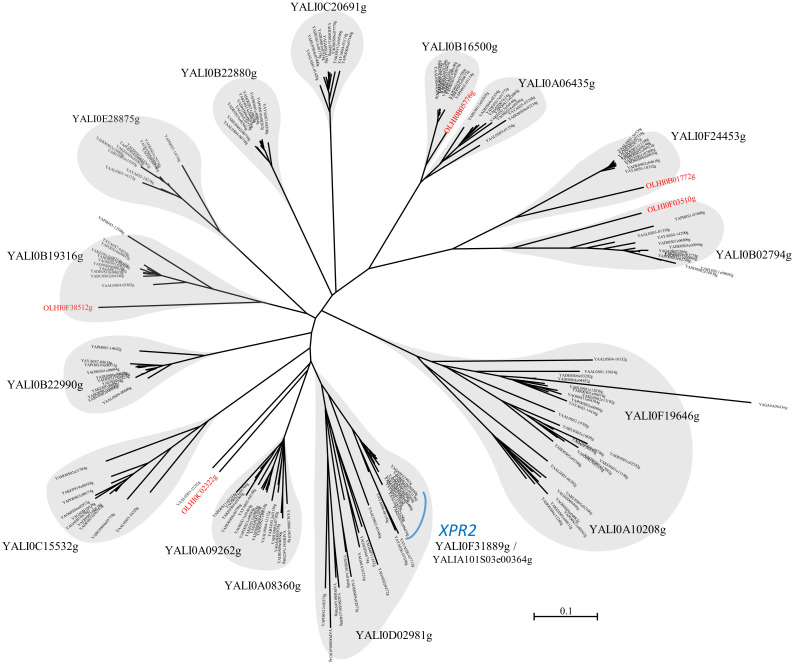
Unrooted circular phylogenetic tree showing the relationship between the alkaline proteases found in the genomes of *Yarrowia* clade species. The thirteen groups of proteins are depicted by grey clouds. The tree was constructed with phyML with a LG evolutionary model. Proteins closely related to *XPR2* are highlighted by a blue curve. Protein-coding genes of *Y. lipolytica* are in large characters, and those of *C. hispaniensis* are in red.

**Figure 2 ijms-24-06514-f002:**
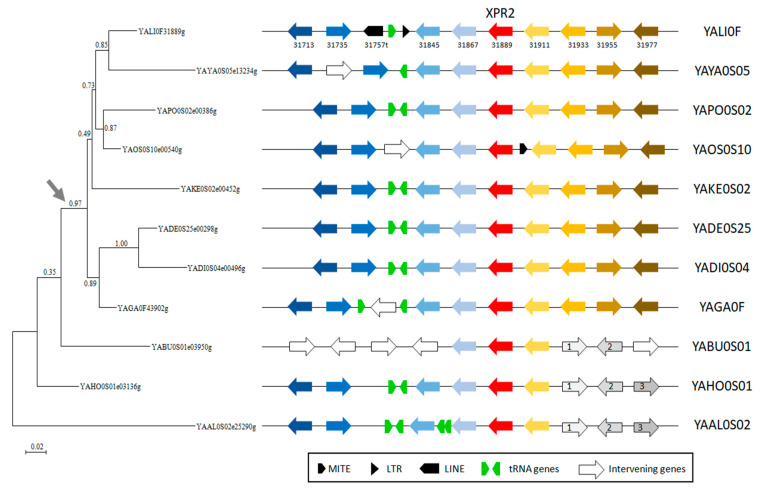
Synteny analysis of alkaline extracellular protease *XPR2* from different *Yarrowia* species. The phylogenetic tree was constructed from an alignment of 446 amino acids with phyML with a LG evolutionary model. *XPR2* orthologs are in red and flanking genes are in different blue or yellow to brown colors reflecting orthologs in the different species. Transposable elements (MITE, LTR, LINE) and tRNA genes are represented as described in the legend. The grey arrow close to the phylogenetic tree indicates the rearrangement event leading to the actual gene order in *Y. lipolytica* and closely related species.

**Figure 3 ijms-24-06514-f003:**
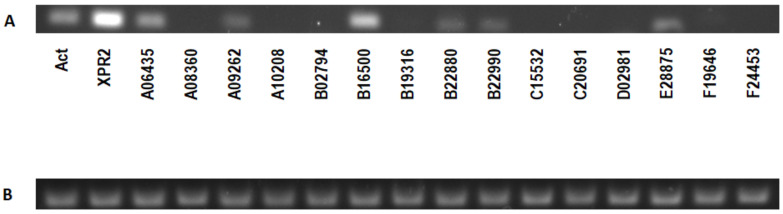
(**A**) Expression of alkaline proteases in *Y. lipolytica* W29 growing on BSG medium, (**B**) Verification of primers on genomic DNA. The numbers correspond to the analyzed genes, *XPR2* refers to the alkaline extracellular protease, and *Act* refers to actin gene.

**Figure 4 ijms-24-06514-f004:**
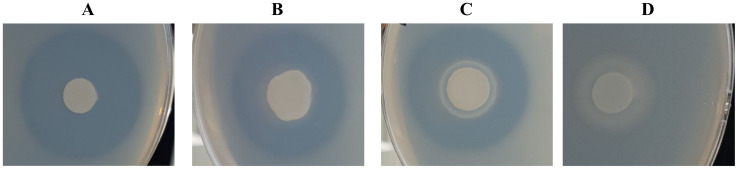
Proteolysis zone observed on the skimmed milk agar plate culture of (**A**) *Y. yakushimensis*, (**B**) *Y. keelungensis*, (**C**) *Y. alimentaria* presented strong proteolytic properties, and (**D**) *Y. brassicae* of the lower activity.

**Figure 5 ijms-24-06514-f005:**
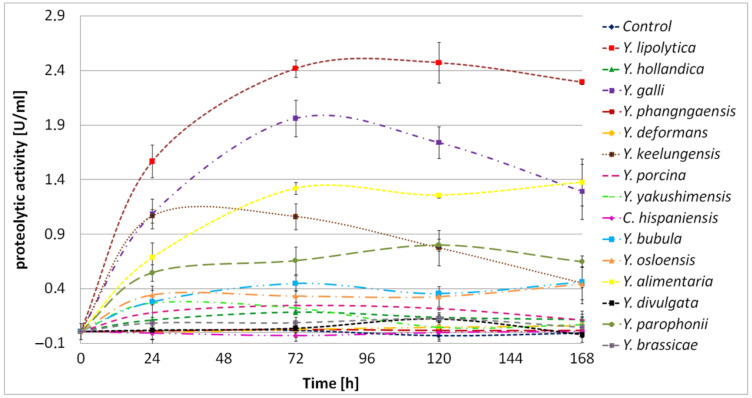
Proteolytic activity of *Yarrowia* clade species grown in BSG protein medium as a function of time. The results are presented as mean values ± SD. *n* = 3.

**Figure 6 ijms-24-06514-f006:**
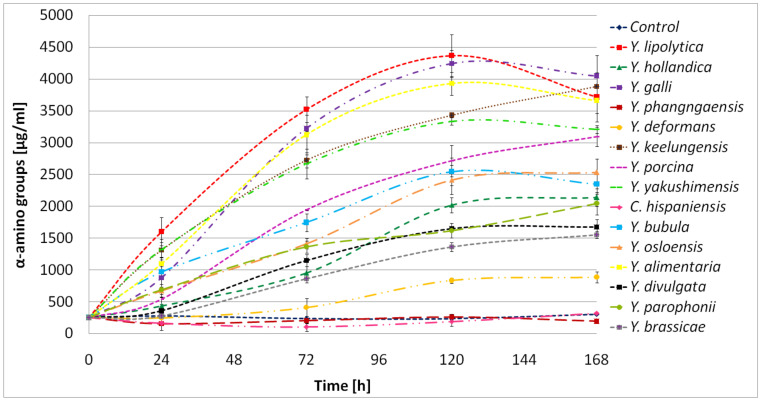
Free α-amino groups concentration in *Yarrowia* clade cultures grown in BSG protein medium as a function of time. The results are presented as mean values ± SD. *n* = 3.

**Figure 7 ijms-24-06514-f007:**
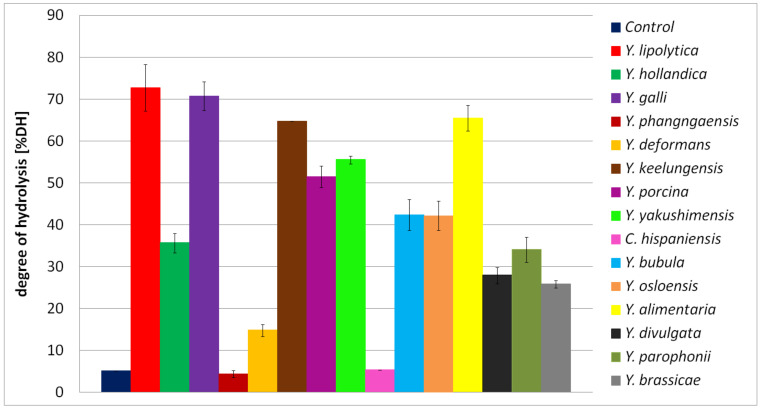
Degree of BSG protein hydrolysis (DH%) in the culture of *Yarrowia* clade species. The results are presented as mean values ± SD. *n* = 3.

**Figure 8 ijms-24-06514-f008:**
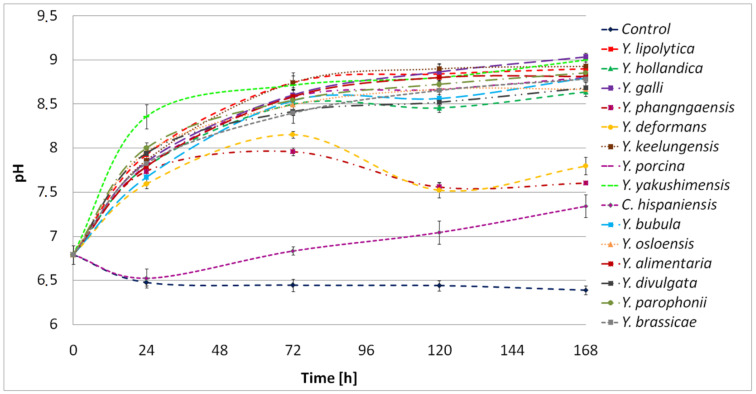
Kinetics of pH in the *Yarrowia* clade culture performed in BSG medium as a function of time. The results are presented as mean values ± SD. *n* = 3.

**Figure 9 ijms-24-06514-f009:**
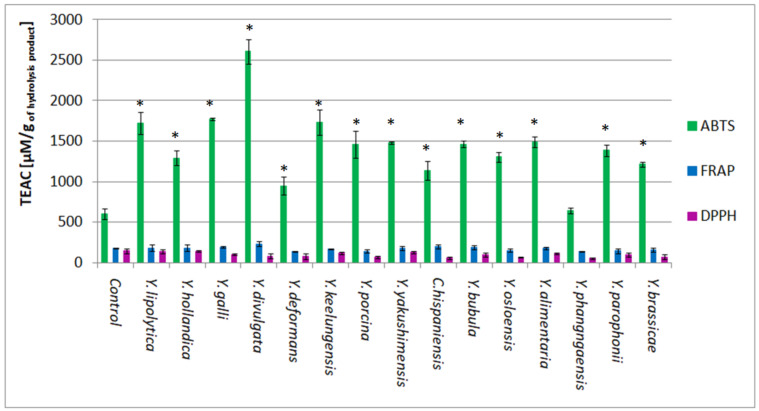
Comparison of antioxidant activity of BSG protein hydrolysates obtained in different yeast species cultures belonging to the *Yarrowia* clade. The results are presented as mean values ± SD. *n* = 3. The significance level was set at * *p* < 0.05.

**Table 1 ijms-24-06514-t001:** Number of alkaline proteases identified in the genomes of *Yarrowia* clade species.

No.	Species and Strain	Number of Alkaline Protease
1	*Y. lipolytica* A101	15
2	*Y. lipolytica* E150 (CLIB 122)	16
3	*Y. lipolytica* H222	16
4	*Y.alimentaria* CBS 10151	19
5	*Y. bubula* CBS 12934	16
6	*Y. deformans* CBS 2071	15
7	*Y. divulgata* CBS 11013	15
8	*Y. galli* CBS 9722	21
9	*Y. hollandica* CBS 4855	15
10	*Y. keelungensis* CBS 11062	17
11	*Y. osloensis* CBS 10146	19
12	*Y. phangngaensis* CBS 10407	5
13	*Y. porcina* CBS 12935	19
14	*Y. yakushimensis* CBS 10254	15
15	*C. hispaniensis* CBS 9996	5

**Table 2 ijms-24-06514-t002:** Predictable expression of the alkaline proteases of *Y. lipolytica* in various culture conditions and prediction of putative signal peptide.

YALI Protease	YNB			Expression YPD	Expression on YPD at pH 4	Functional Annotation	Signal Peptide with SignalP-5.0	Signal Peptide with DeepLoc-1.0	Type of Signal Sequence
	Rank of Expression OA	Rank of Expression GLU	Rank of Expression TB						
YALI0A06435g	941	3795	4250	moderate	low	similar to uniprot|P09232 *S. cerevisiae* YEL060C (ohnolog of YOR003W) PRB1 protease B, vacuolar	0.9911	0.9966	pre-pro?
YALI0A08360g	3831	3360	5852	low	low	weakly similar to uniprot|P09230 *Y. lipolytica XPR2* Alkaline extracellular protease precursor	0.9894	0.999	pre-pro?
YALI0A09262g	6026	6038	6189	low	low	uniprot|F2Z604/Q6H9N6 *Y. lipolytica* YALI0A09262g AEP2 Alkaline extracellular protease precursor	0.9858	0.9988	pre-pro?
YALI0A10208g	NE	NE	NE	NE	NE	similar to uniprot|P09230 *Y. lipolytica XPR2* Alkaline extracellular protease precursor	0.996	0.9989	
YALI0B02794g	5431	5513	5001	low	low	similar to uniprot|P25381 *S. cerevisiae* YCR045C Probable subtilisin-family protease with a role in the formation of the dityrosine layer of the spore	0.9983	0.9914	
YALI0B16500g	213	308	246	high	high	similar to uniprot|P09232 *S. cerevisiae* YEL060C (ohnolog of YOR003W) PRB1 protease B, vacuolar	0.9068	0.8928	pre-pro
YALI0B19316g	6162	6181	5646	low	low	similar to uniprot|P09230 *Y. lipolytica XPR2* Alkaline extracellular protease precursor	0.9192	0.9999	pre-pro
YALI0B22880g	5772	5794	5898	low	low	similar to uniprot|P09230 *Y. lipolytica XPR2* Alkaline extracellular protease precursor	0.9985	0.9992	pre-pro
YALI0B22990g	5953	5838	6124	low	low	weakly similar to uniprot|P09230 *Y. lipolytica XPR2* Alkaline extracellular protease precursor	0.9942	0.997	
YALI0C15532g	NE	NE	NE	NE	NE	some similarities with uniprot|P09230 *Y. lipolytica XPR2* Alkaline extracellular protease precursor	0.9897	0.6785	pre-pro
YALI0C20691g	5213	5520	5477	low	low	similar to uniprot|P09230 *Y. lipolytica XPR2* Alkaline extracellular protease precursor	0.9912	0.9998	pre-pro
YALI0D02981g	5469	5561	5850	low	low	highly similar to uniprot|P09230 *Y. lipolytica XPR2* Alkaline extracellular protease precursor	0.98	0.9994	degenerate pre-pro
YALI0E28875g	5377	5182	5176	low	low	similar to uniprot|P09230 *Y. lipolytica XPR2* Alkaline extracellular protease precursor	0.9668	0.9995	
YALI0F19646g	6201	6207	6117	NE	NE	weakly similar to uniprot|P58371 *Magnaporthe grisea* Subtilisin-like proteinase Spm1 precursor	0.9939	0.9997	degenerate pre-pro
YALI0F24453g	5989	6046	5675	low	low	weakly similar to uniprot|P25381 *S. cerevisiae* YCR045c similarity to serin proteases	0.9916	0.9996	
*XPR2*	5263	22	5880	moderate	low	uniprot|P09230 *Y. lipolytica* Alkaline extracellular protease	0.9833	0.9995	degenerate pre-pro

Expression on YNB medium with OA, GLU, and TB was performed with W29 [26], whereas expression on YPD and YPD at pH 4 was performed with a derivative of E150 (CLIB 122) of genotype E150, *ku70::URA3*, *LEU2* (Neuvéglise, unpublished). OA—oleic acid; GLU—glucose; TB—tributyrin.

**Table 3 ijms-24-06514-t003:** Proteolysis zone diameter of *Yarrowia* clade species growing on skimmed milk agar plates. The results are presented as mean values ± SD. *n* = 3.

Tested Species	Control	*Y. lipolytica*	*Y. hollandica*	*Y. galli*	*Y. phangngaensis*	*Y. deformans*	*Y. keelungensis*	*Y. porcina*
Proteolytic zone diameter [mm]	-	14.80 (±0.40)	8.83 (±0.76)	11.14 (±1.97)	-	-	14.02 (±0.04)	12.90 (±0.86)
**Tested species**	** *Y. yakushimensis* **	** *C. hispaniensis* **	** *Y. bubula* **	** *Y. osloensis* **	** *Y. alimentaria* **	** *Y. divulgata* **	** *Y. parophonii* **	** *Y. brassicae* **
Proteolytic zone diameter [mm]	15.05 (±0.52)	-	9.18 (±0.32)	9.05 (±0.08)	13.51 (±1.01)	9.52 (±1.32)	9.73 (±0.64)	8.33 (±1.52)

**Table 4 ijms-24-06514-t004:** Molecular masses of *Yarrowia* clade enzymes observed on gelatin gel.

Host	Molecular Mass [kDa]										Reference to Appendix A
	30	35	37–39	40	55	65	100	110	130	180	
*Y. lipolytica*								×	×		9A, lane 3
*Y. alimentaria*								×	×		9A, lane 6
*Y. hollandica*					×				×		9C, lane 11
*Y. divulgata*									×	×	9B, lane 10
*Y. brassicae*									×	×	9B, lane 8
*Y. porcina*	×									×	9C, lane 12
*Y. keelungensis*	×										9A, lane 5
*Y. galli*			×	×							9A, lane 4
*Y. parophonii*			×	×							9B, lane 7
*Y. osloensis*				×		×	×				9C, lane 13
*Y. bubula*							×				9B, lane 9
*Y. yakushimensis*		×									9C, lane 14

**Table 5 ijms-24-06514-t005:** Primers used for alkaline proteases expression analysis.

No	Gene	Primer Name	Sequence
1	YALI0F31889	qXPR2-F	AATCCAGAAGCTGGACTCTC
		qXPR2-R	CTCGGGGAGAGACACAATG
2	YALI0A06435	qA06435-F	TGCTTGGAAGCTGCGATATG
		qA06435-R	AGGGAGCGTCCAGTTCTTC
3	YALI0A08360	qA08360-F	CATCTTCAAGGACGGTGTTTC
		qA08360-R	TGGTGACCAAGTCGGAGAAC
4	YALI0A09262	qA09262-F	GTATCAAGACCGCCTTTGAC
		qA09262-R	GTACCCTGGTACACGTTCTG
5	YALI0A10208	qA10208-F	CCAACAATACTGCAGGAAACG
		qA10208-R	AAATACGAGCCAGGTGATAGG
6	YALI0B02794	qB02794-F	CGATCGACTCTGGCATTC
		qB02794-R	TTCGACAGCGAAATCACCTC
7	YALI0B16500	qB16500-F	CTCACGAGTTGTGGGTTGAG
		qB16500-R	GTAGGCAATGTCCTTGTGTC
8	YALI0B19316	qB19316-F	GGATACCACGGGCTGTTTG
		qB19316-R	CCGGTTGACCTGGTTATCTC
9	YALI0B22880	qB22880-F	GTCGACGCGTTCACTGATG
		qB22880-R	ACTGGAAGTCTCGCTTGTG
10	YALI0B22990	qB22990-F	CGAGTCTGCTGTTGACATTG
		qB22990-R	ACCCGAATATTTTGAAGCCATC
11	YALI0C15532	qC15532-F	AAGGCATCACTGCTGACTTC
		qC15532-R	GGGAGCTGATCACAGTAGAC
12	YALI0C20691	qC20691-F	CTGACCTCGACTCTGACATC
		qC20691-R	AAAGGCATTGAGAGCATGTG
13	YALI0D02981	qD02981-F	CGAAGAGGACCGGTACATTG
		qD02981-R	GGTTCACGATGGTGGAGTTG
14	YALI0E28875	qE28875-F	AGACTCGAGAGGAGCACTAC
		qE28875-R	CTGTGTCGGCAGAGAACTTG
15	YALI0F19646	qF19646-F	ACGTCGGAACCTTTTCCAAC
		qF19646-R	GGGTGTTTCGAGGGTTCTG
16	YALI0F24453	qF24453-F	TGCGTTCGGCAAGTTTAAAG
		qF24453-R	GGACAGTTCCAGAGACATCATC
17	YALI0D08272	qAct1-F	CAGTCCAAGCGAGGTATCC
		qAct1-R	TCTTCTCTCGGTTGGACTTG
18	YADE0S25e00298g1_1	qDE-00298-F	TCGCCGGAAAGTTCAACTC
		qDE-00298-R	TAGGCGTAGCTGCTTGC
19	YADE0S07e01684g1_1	qDE-Act1-F	GCACCACACCTTCTACAAC
		qDE-Act1-R	GAGAAACCGGAGTAGATGGG

## Data Availability

The data presented in this study are available on request from the corresponding author.

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
