# Peer review of "Comparative Analysis of the Alkaline Proteolytic Enzymes of Yarrowia Clade Species and Their Putative Applications"

_ijms, 2023, doi:10.3390/ijms24076514_

Round 1
Reviewer 1 Report
In the manuscript of “Comparative analysis of the alkaline proteolytic enzymes of Yarrowia clade species and their putative applications”, the interesting data has been showed. The proteolytic enzymes of Yarrowia clade were analyzed from the gene to potential application. The study compared the alkaline proteases of Yarrowia, and it is meaningful. There are some problems in the manuscript.
1. the comparison of proteolytic enzymes of secreted by Yarrowia, based on the table 1, there are several number of alkaline protease in each strain. How to make sure that the band, according to the gelatin zymogram, was formed by one protease? One strain might secrete over one protease when it was cultured in a medium.
2. The comparison was meaningful one by one protease, however, in the study, they were mixed of proteases. They might be defined by mRNA of proteases.
3. Proteases have the specific of substrates and the optimal pH and temperature of reaction, so, the results of comparison might lead to misunderstand the application.
4. Figure 5, the result might be changed if the culture temperature, pH, ion concentration…..were changed.
5. The antioxidant activity of BSG protein hydrolysates depends on several factors: the components of the peptides, oligopeptides, oligosaccharides. Moreover, the kind of amino acid was exposed from proteins after hydrolysis, the kind of oligosaccharides. So, they could be discussed.
6. The figures of bars and lines need to reconstructed.
Author Response
Reviewer 1
In the manuscript of “Comparative analysis of the alkaline proteolytic enzymes of Yarrowia clade species and their putative applications”, the interesting data has been showed. The proteolytic enzymes of Yarrowia clade were analyzed from the gene to potential application. The study compared the alkaline proteases of Yarrowia, and it is meaningful. There are some problems in the manuscript.
- the comparison of proteolytic enzymes of secreted by Yarrowia, based on the table 1, there are several number of alkaline protease in each strain. How to make sure that the band, according to the gelatin zymogram, was formed by one protease? One strain might secrete over one protease when it was cultured in a medium.
Response: In gelatin zymography and SDS-PAGE electrophoresis the rate of compounds migration depends on the molecular weight. In the performed research, one band observed on the zymogram gel correspond to the one band on SDS-PAGE gel. In studies where SDS-PAGE is used, it is standardly assumed that one band corresponds to one compound. However, the presence of two proteases of the same molecular weight in the sample is unlikely, but as you noticed this cannot be excluded. However, based on the expression analysis done for Y. lipolytica, which was presented in the manuscript, one can hypothesize that most of the proteases are not expressed, and one band really corresponds to one protein. Also, the molecular weight calculation showed differences in the molecular mass of the proteins which were expressed, e.g.: XPR2 having 46,9 kDa whereas YALI0B16500 having 54,6 kDa. We are of course aware that some posttranslational modification may occur which will change the molecular weight, however, after checking for example N-glycosylation, both, XPR2 and YALI0B16500 shows one glycosylation site being occupied by N-glycans what suggests, that they will still differ in the molecular weight.
It is not possible to make sure that one band observed on the zymogram gel was formed by one protease. Based on the amino acid sequence analysis, proteases secreted by Yarrowia clad are characterized by similar molecular weight. For example the molecular mass of Y. lipolytica proteases is in the range of 40 – 60kDa, except the protease encoded by YALI0C15532g of the molecular mass 91kDa. Therefore, good separation of proteases is not possible under applied conditions. In addition, due to the lack of reducing agents, such as β-mercaptoethanol (BME) or dithiothreitol (DTE) some observed bands correspond to the dimeric form of enzymes.
- The comparison was meaningful one by one protease, however, in the study, they were mixed of proteases. They might be defined by mRNA of proteases.
Response: In this work we wanted to prepare a cocktail of Yarrowia clade proteases from each species. The basic bioinformatics work, was analyzing each possible protease and their further sequence comparison, etc. However, in the culture on BSG, only some proteases were expressed, as shown for example for Y. lipolytica (Fig. 3). This analysis indicated that only some of the proteases were expressed, with the XPR2 being the one with the highest expression.
- Proteases have the specific of substrates and the optimal pH and temperature of reaction, so, the results of comparison might lead to misunderstand the application.
Response: In the future perspective of using the proteases commercially we would like to develop a process of bioactive peptides production from the BSG proteins using Yarrowia clade proteases (line 868 - 872). The performed studies showed, that the BSG medium is suitable for inducing the secretion of proteases, as well as providing conditions for their high activity. Furthermore, the analyzed peptides showed some interesting properties (e.g. antioxidant), so the concept we propose seems to be correct. Furthermore, we do not want to purify and obtain pure proteases. Our goal is to prepare a raw protease cocktail, which will show activity in one common buffer at the same temperature.
- Figure 5, the result might be changed if the culture temperature, pH, ion concentration…..were changed.
Response: We agree with the Reviewer opinion. The presented results refer to the conditions described in the Materials and Methods section.
- The antioxidant activity of BSG protein hydrolysates depends on several factors: the components of the peptides, oligopeptides, oligosaccharides. Moreover, the kind of amino acid was exposed from proteins after hydrolysis, the kind of oligosaccharides. So, they could be discussed.
Response: We fully agree with the Reviewer’s opinion, therefore, we briefly discussed the impact of the other components of BSG on antioxidant activity of the hydrolysates (lines 515-522). Thank you for that valuable comment.
- The figures of bars and lines need to reconstructed.
Response: All figures in the manuscript have been prepared in a way that allows us to present our results in the best possible way. Due to the large number of results being a consequence of the study of 15 yeast strains, the charts are complex. However, all the results we obtain are worth of publishing and any attempt to limit the data will negatively affect the scientific value of the Manuscript.
Reviewer 2 Report
The manuscript is informative, but there were some critical things to correct, which must be improved by correcting.
Overall, the codes expressed in the manuscript do not match, which needs to coincide with helping to understand readers.
As a representative example, using of XPR2 and YALI0F31845g are mixed, which makes the reader confused easily.
The use of ',' and '.' is not unified (e.g., 14,02 in Table 3). The author needs to unify the use of the two punctuation marks. In addition, the manuscript did not match the spacing between numbers and units, which seems necessary to correct and match.
The way of representing figures should be unified in the manuscript. For example, there were differences marked for figures which were expressed, such as Figure 1 and Fig. 4.
When the abbreviation first came out, the author had to use the abbreviation after writing the full name. However, In the manuscript, those were the abbreviation in Section 2 and wrote the full name was it in Section 3.
Line 162: ~duplication of the ancestral copy, YAYA0S05e13234g in another chromosome,~
it seemed difficult to find the information about YAYA0S05e13234g that was referred to in Figure S1 or Figure 2. The notation should be improved.
Line 166: ~or XPR2 and YALI0F31845g homologues with a complex rearrangement~
As expressed, it is not easy to find the information when looking for information on YALI0F31845 in Figure S1. It is YlXPR2 on the phylogenetic tree, and it is presumed to refer to the F31845 gene in the correct figure of the phylogenetic tree. However, it is difficult for the reader to recognize it intuitively.
Line 206~207: ~. In turn, no expression of XPR2 was observed on YPD glucose at pH 4 (Table 2).~
Table 2 of the manuscript described that YAL10F31889g, XPR2, is not expressed under YPD glucose at pH 4. However, according to Table 2, the degree of expression is "low" under the condition. The manuscript needs to revise the part where the text and the table do not match.
Table 2: The part expressed about expression needs to be modified.
When the manuscript expressed the expression level in the YNB condition, it was expressed using color. However, the manuscript did not explain how the colors and numbers expressed in the table correlate. it is necessary to explain how the two expressions are correlated.
In addition, unlike the YNB condition, the manuscript described the YPD condition as high, moderate, low, and NE. Why did the YPD condition express a difference from the YNB condition?
Figure 3: Missing description in illustration.
The manuscript should have explained what Act is. There needed to be a description of the Act in the text too. However, from Table 5, which comes out later, it can be estimated that what Act refers to is YALI0D08272.
It would be better to add an explanation for this part.
Line 248~250: ~. The highest proteolytic activity were ~ (Fig.4).
It would be better to write down only Y. yakushimensis that showed the highest activity or list it in the order of the highest activity. In addition, there are no photos of Y. yakusimensis, which showed the highest activity among the strains listed in the text in Fig. 4, so the author must add a photo or explain why the photo is missing from the text.
Fig. 4: Like the previous figures, 'Fig. 4' should be corrected to 'Figure 4'.
The description of the picture does not match the picture. The manuscript described that C and D have low activity, but in the reader's view, the activity of A, B, and C, excluding D, appears to be high. Readers may wonder at what point measured and what the basis is for judging whether the activity is low or high.
Figure 5: Incorrect expression needs to be corrected
Please, Change ',' to '.' in the legend representation of the y-axis to indicate it.
It is hard to distinguish between each piece of data in the graph (the same goes for Fig. 6 and Fig. 7).
Line 333~337: ~the concentration of α-amino groups reach ~, respectively.
The values are the maximum concentration for each strain.
If the author wants to show the maximum concentration per strain rather than at a specific time (e.g., a concentration at 120 hours)
Section 2.6 Profile of Yarrowia clade proteolytic enzymes secreted in BGS based medium
(1) Line 399~405:
According to the results, strains with high protein activity (Y. lipolytica and Y. alimentaria) and strains with low protein activity (Y. divulgata and Y. brassicae) have 130 kDa of protein at the same time. The manuscript argues that 110 kDa of protein, of which only two strains (Y. lipolytica and Y. alimentaria) have high protein activity, plays an essential role in BSG protein hydrolysis.
This does not explain in previous experimental results that Y. galli, like Y. lipolytica, does not have a protein of 110 kDa despite high BSG protein hydrolysis.
(2) Line 417~425:
In Fig. 6, Y. porcina showed a free-amino group concentration of 3000 μg/mL or more; in Fig. 7, the hydrolysis degree of DH was 50%. However, it is weird that the manuscript expresses Y. porcina as a low protein decomposition. The basis for judging by low protein decomposition should be described.
Line 505~508: ~. It is likely related to the degree of hydrolysis (DH%) of BSG protein, ~.
Y. divulgata shown in Figure 9, did not show high activity in the previous results. In Figure 7, strains with a value of 25% DH or more appear to have a value of 1300 μM TEAC/g or more due to ABTS.
In the case of Y. divulgata, the pattern is different from the previous experimental results, so it would be good for the author to add an explanation.
Conclusions: Additional content about the putative application included in the title of the manuscript is needed.
Author Response
Reviewer 2
The manuscript is informative, but there were some critical things to correct, which must be improved by correcting.
Overall, the codes expressed in the manuscript do not match, which needs to coincide with helping to understand readers.
Response: Thank you for the valuable suggestion. The whole document has been checked and corrected accordingly.
As a representative example, using of XPR2 and YALI0F31845g are mixed, which makes the reader confused easily.
Response: The document has been standardized in the context of the codes appearing in the manuscript. Therefore XPR2 is used instead YALI0F31889g. Moreover, the correlation between Aep, XPR2 and YALI0F31889g have been introduced (line 160).
The use of ',' and '.' is not unified (e.g., 14,02 in Table 3). The author needs to unify the use of the two punctuation marks. In addition, the manuscript did not match the spacing between numbers and units, which seems necessary to correct and match.
Response: Thank you for the detailed analysis. The entire document has been checked and the use of ',' and '.' as well as the lack of spacing between numbers and units was corrected.
The way of representing figures should be unified in the manuscript. For example, there were differences marked for figures which were expressed, such as Figure 1 and Fig. 4.
Response: Thank you once again for the valuable comment. Differences in the method of marking Figures have been unified (line 268).
When the abbreviation first came out, the author had to use the abbreviation after writing the full name. However, In the manuscript, those were the abbreviation in Section 2 and wrote the full name was it in Section 3.
Response: The whole document was corrected regarding the use of abbreviations after the full name. We hope there are no more oversights.
Line 162: ~duplication of the ancestral copy, YAYA0S05e13234g in another chromosome,~
it seemed difficult to find the information about YAYA0S05e13234g that was referred to in Figure S1 or Figure 2. The notation should be improved.
Response: Indeed, the nomenclature was not homogenous for all species. It has been changed in Fig. 2 and Fig. S1 to follow the same rule. The four first digits refer to the species name “YAYA”, “0” indicates the first version, “S5” indicates the scaffold number, “e” for (genetic) element, and the last 5 digits indicate the number of the gene.
The paragraph about the successive duplications that occurred in Y. yakushimensis has been simplified. It should be more readable now (lines 173-180).
Line 166: ~or XPR2 and YALI0F31845g homologues with a complex rearrangement~
As expressed, it is not easy to find the information when looking for information on YALI0F31845 in Figure S1. It is YlXPR2 on the phylogenetic tree, and it is presumed to refer to the F31845 gene in the correct figure of the phylogenetic tree. However, it is difficult for the reader to recognize it intuitively.
Response: We agree that this paragraph was difficult to understand in its previous form. It has been simplified, and YlXPR2 has been changed to XPR2.
Line 206~207: ~. In turn, no expression of XPR2 was observed on YPD glucose at pH 4 (Table 2).~
Table 2 of the manuscript described that YAL10F31889g, XPR2, is not expressed under YPD glucose at pH 4. However, according to Table 2, the degree of expression is "low" under the condition. The manuscript needs to revise the part where the text and the table do not match.
Response: Thank you for your accurate suggestion. The text of the manuscript has been corrected to match the data in the Table 2 (line 221).
Table 2: The part expressed about expression needs to be modified.
When the manuscript expressed the expression level in the YNB condition, it was expressed using color. However, the manuscript did not explain how the colors and numbers expressed in the table correlate. it is necessary to explain how the two expressions are correlated.
In addition, unlike the YNB condition, the manuscript described the YPD condition as high, moderate, low, and NE. Why did the YPD condition express a difference from the YNB condition?
Response: The colors were used to have a first look at the differential expression of the genes. It has been removed in the revised version.
The data on the YNB conditions have been previously published from RNA-seq data, while those on YPD have not been and were extracted from a cDNA sequence analysis. As the number of cDNA is much smaller than that of the RNA-seq data, it was not possible to do the same type of statistical analysis, only to give a trend. A sentence has been added in Material and method (line 648).
Figure 3: Missing description in illustration.
The manuscript should have explained what Act is. There needed to be a description of the Act in the text too. However, from Table 5, which comes out later, it can be estimated that what Act refers to is YALI0D08272.
It would be better to add an explanation for this part.
Response: As suggested by the Reviewer, Act was explained both, in the Materials and Methods (lines 668-669), as well as below the Fig. 3.
Line 248~250: ~. The highest proteolytic activity were ~ (Fig.4).
It would be better to write down only Y. yakushimensis that showed the highest activity or list it in the order of the highest activity. In addition, there are no photos of Y. yakusimensis, which showed the highest activity among the strains listed in the text in Fig. 4, so the author must add a photo or explain why the photo is missing from the text.
Response: The Yarrowia species were listed in the order of the highest activity (lines 265-267). A photo of Y. yakushimensis was added (Figure 4).
Fig. 4: Like the previous figures, 'Fig. 4' should be corrected to 'Figure 4'.
Response: 'Fig. 4' has been corrected to 'Figure 4'.
The description of the picture does not match the picture. The manuscript described that C and D have low activity, but in the reader's view, the activity of A, B, and C, excluding D, appears to be high. Readers may wonder at what point measured and what the basis is for judging whether the activity is low or high.
Response: The description of the Figure 4 was corrected. We also decided to add the photo of Y. yakushimensis of the highest activity (according to the Reviewer’s suggestion) and remove the photo of Y. bubula to facilitate the correct reception of this figure.
Figure 5: Incorrect expression needs to be corrected
Please, Change ',' to '.' in the legend representation of the y-axis to indicate it.
It is hard to distinguish between each piece of data in the graph (the same goes for Fig. 6 and Fig. 7).
Response: The legend representation of the y-axis has been changed in order to replace ',' with '.' In the studies 15 different strains of Yarrowia were tested. We believe that all the data presented in the graphs are important and constitute the basis for future research, for each of the strains. Therefore, we don't want to remove any of them. The graphs were constructed using different colors and line styles, as well as different tags. We believe that this form of presenting results is the best possible.
Line 333~337: ~the concentration of α-amino groups reach ~, respectively.
The values are the maximum concentration for each strain.
If the author wants to show the maximum concentration per strain rather than at a specific time (e.g., a concentration at 120 hours)
Response: The values presented in line 344 are the maximum concentration of free amino groups, respectively for Y. lipolytica, Y. galli, Y. alimentaria and Y. keelungensis. These are not values for a specific point in time. However, bearing in mind The Reviewer’s opinion, the text below has been corrected (line 362).
Section 2.6 Profile of Yarrowia clade proteolytic enzymes secreted in BGS based medium
(1) Line 399~405:
According to the results, strains with high protein activity (Y. lipolytica and Y. alimentaria) and strains with low protein activity (Y. divulgata and Y. brassicae) have 130 kDa of protein at the same time. The manuscript argues that 110 kDa of protein, of which only two strains (Y. lipolytica and Y. alimentaria) have high protein activity, plays an essential role in BSG protein hydrolysis.
This does not explain in previous experimental results that Y. galli, like Y. lipolytica, does not have a protein of 110 kDa despite high BSG protein hydrolysis.
Response: Gelatin zymography revealed enzymes involved in the hydrolysis of BSG proteins. Therefore, in the Y. galli culture medium an enzymes with a molecular weight of 37-39 kDa and 40 kDa are responsible for BSG protein hydrolysis (Figure S2.). According to Table 1, in the genome of Y. galli there is 21 genes encoding alkaline proteases of which only two are active in the BSG medium. Based on amino acid sequence analysis, the molecular weight of Y. lipolytica protease is in the range of 40-60 kDa, except the protease of the mass of 91 kDa encoded by YALI0C15532g. Therefore, observed bands at 110 kDa, 130 kDa represent enzyme complexes. We supplemented the manuscript with information on enzyme complexes (lines 403 – 409) and corrected the result part.
(2) Line 417~425:
In Fig. 6, Y. porcina showed a free-amino group concentration of 3000 μg/mL or more; in Fig. 7, the hydrolysis degree of DH was 50%. However, it is weird that the manuscript expresses Y. porcina as a low protein decomposition. The basis for judging by low protein decomposition should be described.
Response: We totally agree with the Reviewer’s opinion. Therefore, the manuscript has been revised accordingly (lines 307–308; 438-439).
Line 505~508: ~. It is likely related to the degree of hydrolysis (DH%) of BSG protein, ~.
- divulgata shown in Figure 9, did not show high activity in the previous results. In Figure 7, strains with a value of 25% DH or more appear to have a value of 1300 μM TEAC/g or more due to ABTS.
In the case of Y. divulgata, the pattern is different from the previous experimental results, so it would be good for the author to add an explanation.
Response: We are very thankful for this suggestion. In the manuscript, lines 515-527, we have undertaken to explain different pattern observed in Y. divulgata culture.
Conclusions: Additional content about the putative application included in the title of the manuscript is needed.
Response: Paragraphs describing both the expected applications of Yarrowia clade proteases and antioxidant peptides resulting from BSG proteolysis have been added to the manuscript (lines 578 – 596; 856 – 872).
Reviewer 3 Report
In the reviewed original research article, Ciurko et al. comprehensively analyse alkaline proteolytic enzymes from different Yarrowia clade species. The novelty of the research concerns the fact that not only enzymes from Yarrowia lipolytica but various Yarrowia strains were analysed.
As a general remark, the article is professionally written, and the overall appearance is quite satisfactory. The work fits the journal's scope well and might be of interest to a broad readership of the journal. Nonetheless, some questions and unclarities arose while reviewing the manuscript, which should be addressed and/or answered by the authors before the article might be accepted (minor revision). I hope my comments below are helpful and will contribute to improving the quality of the manuscript.
General comments:
1. The formatting of strain names and units should be checked for the entire document. A new line was started after or before a microbes name in several cases. In some cases, units are not separated from the numerical value by a space. Please check and correct the whole manuscript in this regard.
2. Please check the entire document for the correct use of “BSG” as abbreviation for Brewery spent grain. In some cases, it seems that “BGS” has been used instead.
3. In relation to general comment no. 2: why is the use of BSG as a substrate of industrial or scientific interest? What are the expected market price and availability at an industrial scale? It would be much appreciated if a few sentences would be added to the manuscript addressing and explaining the use of BSG in this work and/or in a broader context.
Specific comments:
1. Lines 415-416: Check the formatting.
2. Lines 470-541: Please explain why the antioxidant capacity is important in this context (future use for industrial application?). Here, please also add a reference value for other commercial antioxidants and compare it to the obtained values of the hydrolysates.
3. Lines 648-672: Here, the procedure for the quantitative determination of the proteolytic activity is described. Why was the assay performed at 55°C, which strongly differs from the cultivation temperature of 25°C? Might this have affected the obtained results? Please comment/explain.
4. Line 763: Change “Ten” to “10”.
5. Lines 776-785: The conclusions section sufficiently summarises the most important finding. Anyhow, it would be much appreciated if the authors would add a few sentences for potential future studies, which might be based on the obtained findings and the anticipated application of the enzymes.
Author Response
Reviewer 3
In the reviewed original research article, Ciurko et al. comprehensively analyse alkaline proteolytic enzymes from different Yarrowia clade species. The novelty of the research concerns the fact that not only enzymes from Yarrowia lipolytica but various Yarrowia strains were analysed.
As a general remark, the article is professionally written, and the overall appearance is quite satisfactory. The work fits the journal's scope well and might be of interest to a broad readership of the journal. Nonetheless, some questions and unclarities arose while reviewing the manuscript, which should be addressed and/or answered by the authors before the article might be accepted (minor revision). I hope my comments below are helpful and will contribute to improving the quality of the manuscript.
General comments:
- The formatting of strain names and units should be checked for the entire document. A new line was started after or before a microbes name in several cases. In some cases, units are not separated from the numerical value by a space. Please check and correct the whole manuscript in this regard.
Response: The entire manuscript has been corrected accordingly. We hope that there are no more errors.
- Please check the entire document for the correct use of “BSG” as abbreviation for Brewery spent grain. In some cases, it seems that “BGS” has been used instead.
Response: The entire document has been checked and the use of “BSG” abbreviation was corrected.
- In relation to general comment no. 2: why is the use of BSG as a substrate of industrial or scientific interest? What are the expected market price and availability at an industrial scale? It would be much appreciated if a few sentences would be added to the manuscript addressing and explaining the use of BSG in this work and/or in a broader context.
Response: We are grateful to the Reviewer for this comment. We decided to add a paragraph explaining these facts in the Introduction (lines 114-123).
Specific comments:
- Lines 415-416: Check the formatting.
Response: Formatting has been corrected.
- Lines 470-541: Please explain why the antioxidant capacity is important in this context (future use for industrial application?). Here, please also add a reference value for other commercial antioxidants and compare it to the obtained values of the hydrolysates.
Response: A paragraph describing the possible future application of peptides generated as the product of BSG hydrolysis have been added (lines 578 – 596).
In work of GrzeÅ›kowiak et al. (2018) the activity of standard antioxidants was analyzed. The activity of glutathione, peptide antioxidant using ABTS assay was estimated to be 1.027 mol TE/mol (3341.68 µM TEAC/g), while using FRAP 0.010 mol TE/mol (32.53 µM TEAC/g). However glutathione antioxidant activity, analyzed by another author was determined to be 196.89 mgTE/g Sample (786,65 µM TEAC/g) and 143.03 mgTE/g Sample (571.45 µM TEAC/g) respectively for ABTS and FRAP assay [2]. Finally glutathione antioxidant activity determined in the work of Gu et al. (2012) reached 2.39 mM Trolox/ mM (7776.65 µM TEAC/g) for ABTS assay. Glutathione antioxidant activity presented by various authors differs significantly. This also applies to literature data of other standard antioxidants. Therefore, the comparison of the antioxidant activity of BSG hydrolysates to that of known antioxidants is difficult to perform.
[1] M. Grzesik, K. NaparÅ‚o, G. Bartosz, I. Sadowska-Bartosz, Antioxidant properties of catechins: Comparison with other antioxidants, Food Chem. 241 (2018) 480–492. https://doi.org/10.1016/j.foodchem.2017.08.117.
[2] G. Luisi, A. Stefanucci, G. Zengin, M.P. Dimmito, A. Mollica, Anti-oxidant and tyrosinase inhibitory in vitro activity of amino acids and small peptides: New hints for the multifaceted treatment of neurologic and metabolic disfunctions, Antioxidants. 8 (2019) 7. https://doi.org/10.3390/antiox8010007.
[3] L. Gu, M. Zhao, W. Li, L. You, J. Wang, H. Wang, J. Ren, Chemical and cellular antioxidant activity of two novel peptides designed based on glutathione structure, Food Chem. Toxicol. 50 (2012) 4085–4091. https://doi.org/10.1016/j.fct.2012.08.028.
- Lines 648-672: Here, the procedure for the quantitative determination of the proteolytic activity is described. Why was the assay performed at 55°C, which strongly differs from the cultivation temperature of 25°C? Might this have affected the obtained results? Please comment/explain.
Response: Normally, the proteolytic activity of enzymes is determined at 55°C. It is considered to be the optimal temperature for proteases activity. It differs significantly from the temperature, that is recognized as optimal for cell growth. Therefore, culture was conducted at 25oC, which we consider optimal for Yarrowia growth, while protease activity was measured at 55°C.
- Line 763: Change “Ten” to “10”.
Response: Changed (line 832).
- Lines 776-785: The conclusions section sufficiently summarises the most important finding. Anyhow, it would be much appreciated if the authors would add a few sentences for potential future studies, which might be based on the obtained findings and the anticipated application of the enzymes.
Response: Thank you for your suggestion, which in our opinion increased the scientific value of the Manuscript. Short paragraph describing potential future studies, which might be based on the obtained results has been added. We also described the expected application of proteolytic enzymes of the Yarrowia clade, as well as biologically active hydrolysis products (line 578–596; 856–872; ).
Round 2
Reviewer 1 Report
For the revised manuscript, the authors have addressed all the questions; therefore, I recommend the manuscript for publication.